# Perspectives on Topical Medical Research in the COVID-19 Era

Michael McAleer [1,2,3,4,5]

1   Department of Finance, Asia University, Wufeng District, Taichung City 41354, Taiwan;
    michael.mcaleer@gmail.com
2   Discipline of Business Analytics, University of Sydney Business School, Sydney, NSW 2006, Australia
3   Econometric Institute, Erasmus School of Economics, Erasmus University Rotterdam,
    3062 PA Rotterdam, The Netherlands
4   Department of Economic Analysis and ICAE, Complutense University of Madrid, 28040 Madrid, Spain
5   Institute of Advanced Sciences, Yokohama National University, Yokohama, Kanagawa 240-8501, Japan

**Abstract:** The SARS-CoV-2 virus that causes COVID-19 has wreaked havoc on the global community in terms of every imaginable parameter. The research output on COVID-19 has been nothing short of phenomenal, especially in the medical and biomedical sciences, where the search for a potential vaccine has been conducted in earnest. Much of the advanced research has been distributed in leading medical journals, including the *Journal of the American Medical Association (JAMA),* where the latest research is distributed on a daily basis. The purpose of this paper is to provide some perspectives on 44 interesting and highly topical research papers that have been published in *JAMA*, at the time of writing, within the past two weeks. The diverse topics include public health, general medicine, internal medicine, oncology, pediatrics, geriatrics, and biostatistics.

**Keywords:** COVID-19; pooling clinical trials; hyperinfection; steroids; treatment; targeted healthcare; population health management; cancer treatment; clinical research; clinical trials; developing vaccines; ranking and rating hospital quality; school closures; interventions for delirium; assessments of COVID-19 death inequities; regulatory safeguards; preventing child abuse and maltreatment; prevalence of healthcare worker burnout; nursing home ratings; challenging oncology practice; addressing racial; ethnic; social and economic divides; violence against sexual minority adolescents; primary tumors; metastasis; stages of cancer; reforming cancer clinical trials; supporting carers; protection and prevention; benign and malignant tumors; reforming cancer clinical trials; protection of healthcare personnel; comparing excess deaths in NYC; 1918 influenza pandemic; the possibility of full recovery from COVID-19; mental health impact of COVID-19 on young adults; ranking and rating nursing home quality; Charles Dickens and cancer outcome disparities in the time of COVID-19; schooling children in the time of COVID-19; duration of isolation after positive COVID-19 test outcomes; stress; anxiety and mental health during COVID-19; ethical clinical trials for a COVID-19 vaccine; oxygenating severe respiratory patients with COVID-19





## Preface

Dear Reader

We have to take this unusual step to start this publication with a *Preface* as the one and sole author of the manuscript, Michael McAleer (1951–2021), has left us during revision. Michael was born on 18th of August 1952 in Yokohama, Japan, as an Australian national. He received his Bachelor's and Master's degrees in Economics from the Monash University, Melbourne, Australia, in 1974 and 1977, respectively, and obtained his doctorate degree in Economics from Queen's University, Canada, in 1981.

Michael was an exceptionally liked and active colleague who, in recent years, served simultaneously at five different institutes, including as University Research Chair Professor (Asia University, Taiwan), Erasmus Visiting Professor of Quantitative Finance (Erasmus

University Rotterdam, The Netherlands), Adjunct Professor (University of Madrid, Spain), Adjunct Professor (University of Canterbury, New Zealand), and IAS Adjunct Professor (Institute of Advanced Sciences, Yokohama National University, Japan). Michael was an elected Distinguished Fellow or Member of several prestigious international societies active in the fields of economics, financial econometrics, sustainability, climate change and risk management, including the International Engineering and Technology Institute (DFIETI), the Academy of the Social Sciences in Australia (FASSA), the International Environmental Modelling and Software Society (FIEMSS), the Modelling and Simulation Society of Australia and New Zealand (FMSSANZ), and the Tinbergen Institute, The Netherlands.

Michael was the author of over 1000 publications and he had over 20,000 citations. He had established a close relationship with MDPI and some of its journals, such as the *Journal of Risk and Financial Management* (*JRFM*), for which he served as Editor-in-Chief between 2013 and 2021 and also guest-edited numerous Special Issues. During the last couple of years, Michael had also become interested in the modern way of publishing in *Sci* (ISSN 2413-4155) and had been an adherent supporter of our open peer review approach.

On the 5th of August 2020, Michael submitted his manuscript entitled "Perspectives on Topical Medical Research in the COVID-19 Era" to our journal, in the hope that it could be reviewed and published soon as a valuable contribution to our scientific community. It is truly tragic that Michael could not complete the reviewing and his revisions on his manuscript as he passed away on the 8th of July this year.

As a mark of respect and in honor of Michael, we have therefore decided to publish his manuscript posthumously and in its original form. Rather than instigating a revision by an outsider and thereby meddling with his original text, we have simply kept his words for us and added the comments of the reviewers for colleagues so they may appraise the scientific content.

We would like to invite our readers to share this rather exceptional manuscript in memory of a rather exceptional colleague, as a valuable piece of science and, first and foremost, as a goodbye to Michael McAleer. Michael, we miss you.
Claus Jacob
*Editor-in-Chief*

## 1. Introduction

The once-in-a-century SARS-CoV-2 virus that causes COVID-19 has wreaked havoc on the global community in terms of every imaginable measure, including healthcare outcomes and costs, financial and economic conditions, and the social fabric, among others.

The research output on COVID-19 has been nothing short of phenomenal, especially in the medical and biomedical sciences, as well as in the social sciences, where the search for a potential vaccine has been conducted in earnest.

Much of the advanced research has been distributed in the leading medical journals, including the *Journal of the American Medical Association (JAMA)*, where the latest research is distributed on a daily basis. In respect of critical evaluation of recent papers on COVID-19, some of which are based on published comments in *JAMA*, [1] McAleer (2020a) discusses one paper in general medicine, [2] McAleer (2020b) evaluates 10 papers in general medicine, internal medicine, and oncology, [3] Chang, McAleer, and Wong (2020) examine 16 papers in general medicine, internal medicine, and oncology, [4] McAleer (2020c) analyzes 15 papers in general medicine, internal medicine, and oncology, and [5] McAleer (2020d) assesses 19 papers in general medicine, global health, healthcare, internal medicine, oncology, and pediatrics.

Many of these papers have been included in the World Health Organization's **WHO COVID-19 Global literature on coronavirus disease** (https://www.who.int/emergencies/diseases/novel-coronavirus-2019/global-research-on-novel-coronavirus-2019-ncov), which is intended to bring "the world's scientists and global health professionals together to

accelerate the research and development process, and develop new norms and standards to contain the spread of the coronavirus pandemic and help care for those affected."

The purpose of this review is to provide some perspectives on 44 interesting and highly topical research papers that have been published in *JAMA*, at the time of writing, mostly within the past two weeks. The diverse topics include public health, general medicine, internal medicine, oncology, pediatrics, geriatrics, and biostatistics.

They are worth highlighting because they cover several highly topical issues in the COVID-19 era, including pooling individual clinical trials to reach fast and effective treatment outcomes for COVID-19, avoiding hyperinfection through the use of steroids in treating and managing the symptoms of COVID-19, targeted future healthcare management for all individuals and institutions involved in population health management in the COVID-19 era, surrogate endpoints in cancer treatment, expectations from COVID-19 clinical research, the safe development of a COVID-19 vaccine, dealing with cancer grief in COVID-19, ranking versus rating hospital quality, effective school closures during COVID-19, reducing exclusions in clinical trial interventions for delirium, weighted versus unweighted assessments of COVID-19 death inequities, strengthening registered COVID-19 clinical trials, unwavering regulatory safeguards in approving vaccines for COVID-19, the prevalence of healthcare worker burnout during COVID-19, nursing home ratings, challenging oncology practice during COVID-19, addressing racial, ethnic, social, and economic divides in COVID-19, violence against sexual minority adolescents, primary tumors, metastasis, stages of cancer, reforming cancer clinical trials, supporting carers, protection and prevention, benign and malignant tumors, reforming cancer clinical trials, the protection of healthcare personnel, comparing excess deaths in NYC, the 1918 influenza pandemic, the possibility of full recovery from COVID-19, the mental health impact of COVID-19 on young adults, ranking and rating nursing home quality, Charles Dickens and cancer outcome disparities in the time of COVID-19, schooling children in the time of COVID-19, the duration of isolation after positive COVID-19 test outcomes, stress, anxiety, and mental health during COVID-19, ethical clinical trials for a COVID-19 vaccine, and oxygenating severe respiratory patients with COVID-19.

## 2. Pooling Individual Clinical Trials for COVID-19

Pool testing for COVID-19 is based on combining a large number of local individual randomized clinical trials (RCTs) from different sites to reach fast and effective treatment outcomes, as compared with a more comprehensive analysis across cities, counties, provinces, states, and countries with vastly larger numbers of patients.

The interesting and informative statistical analysis by [6] Petkova, Antman and Troxel (2020) presents a clear and perceptive viewpoint, which is complementary to [7] Cherif, Grobe, Wang et al., (2020), in the context of limited test availability, or local testing shortages, as well as missing data, all of which are key ingredients in terms of pooling.

It is essential that the pooled treatment outcomes are not too general; otherwise, they will not be pertinent or informative for the local settings from which a large amount of the pooled data have been collected, and hence of less relevance and use for informed healthcare consideration.

An important issue relates to the diversity of the data sets that are being pooled, and determining whether they are dependent or independent in terms of the underlying characteristics of the patients in the local settings.

Arbitrary pooling of individual clinical trials without adequate pre-pooling analysis can lead to excessive sampling variability, and hence to a lack of uniformity in terms of treatment outcomes.

Whether "combining individual patient data from different studies could enable more reliable inferences about the treatment effect estimate than would aggregate data from individual trials" may be problematic, because it depends on the similarity or otherwise aggregating individual clinical trials as a composite as compared with an analysis that is based on aggregated clinical trial data to provide estimates of treatment efficacy.

Uploading minimal subsets of the pooled data on accessible websites would allow different methods and assumptions to be used for statistical modelling purposes, whether based on the cumulative logit approach, Bayesian hierarchies, or otherwise, thereby making empirical evaluations of the treatment outcomes more reliable and robust.

Balancing the issues relating to safety, efficacy, integrity, futility, and harm is critical in protecting the welfare of patients in clinical trials, using appropriate control groups, as well as obtaining optimal medical treatment outcomes based on the most relevant statistical advice.

### 3. Avoiding Hyperinfection through Steroids for COVID-19

The highly informative viewpoint by [8] Stauffer, Alpern, and Walker (2020) on a strategy to avoid any adverse and potentially catastrophic hyperinfection through steroids, however uncommon, in treating and managing the symptoms of COVID-19, is innovative in its conception and presentation.

The balance between treatment and effective outcomes affects the quality of life of patients, especially those who are diagnosed with COVID-19, which may lead to chronic infection.

Information and health warnings should be distributed widely to medical specialists and patients to highlight the prevalence worldwide, especially for countries with significant migration from endemic regions.

Extensive clinical trials, particularly in tropical and subtropical regions, would seem to be warranted to determine the incidence of coinfection and hyperinfection through widely used steroids in treating symptomatic and asymptomatic COVID-19 patients.

### 4. Targeted Future Healthcare Management

The impressive viewpoint on future healthcare management by [9] Grossman, Larson, and Sox (2020) makes it essential reading as a template for all individuals and institutions involved in population health management.

Personalized medicine involves genome-wide DNA sequencing, which might be comprehensible to healthcare professionals, but not to the majority of prospective patients who might require treatment in the future.

Clear rules and regulations are required for an effective and manageable personalized treatment approach, as well as education of the broad treatment population, together with a lot of patience.

Cancer treatment is discussed, but a range of health issues and diagnoses are likely to be covered over time, especially with genetic DNA testing to determine risk factors arising through universal or targeted historical tracing and gene penetrance, and to perform prescriptive diagnostic screening to susceptibility for specific diseases.

Accurate data collected over time from clinical cohort trials are essential for predicting and planning future healthcare management, which will be beneficial to all individuals in a non-discriminatory and affordable healthcare system that is accessible to all who need it.

### 5. Surrogate Endpoints in Cancer Treatment

The important and commendable discussion of using pathologic complete response (pCR) as the endpoints in scientific oncology clinical trials is analyzed sensitively and clearly by [10] Shyr and Syr (2020), with an emphasis on the effective treatment of early-stage breast cancer and prostate cancer.

The difficulty in selecting any surrogate measure lies in assessing its accuracy relative to the effect of treatment on the true endpoint, elimination of symptomatic disease, or reduction in the mortality rate for the individual patient.

Important issues regarding the sample size, namely, the number of patients, drugs, events for treatment, appropriate odds ratios, leading to correlations (linear or otherwise) of the odds and hazard ratios, need to be tackled rigorously from a scientific perspective, with an emphasis on appropriate quantitative techniques.

Alternative methods of validation based on altering the surrogate endpoints and the sample sizes are essential to be able to determine the robustness of the clinical trials, especially with regard to reducing false positive diagnoses.

If the "disease pathways and mechanism of the treatment affect the true clinical outcome and the surrogate end point", assessing the direction of causality would be problematic and misleading.

This key issue needs to be addressed in determining alternative causal relationships between the surrogate and true endpoints.

## 6. Expectations from COVID-19 Clinical Research

The prescient editorial by [11] Salazar, McWilliams, Jr., and Wang (2020) on managed expectations from COVID-19 clinical research shows that, although there has been an impressively large volume of clinical research in the search for a vaccine for COVID-19, the question remains as to how and when the research will lead to safe, effective, timely, available, and accessible treatments for general international distribution.

It is understandable that, in the face of the raging pandemic, there is a maelstrom of active clinical research.

Pool testing for COVID-19 through combining a large number of local individual randomized clinical trials from different sites might lead to fast and effective treatment outcomes (for example, see [7] Cherif, Grobe, Wang et al., (2020), and [6] Petkova, Antman, and Troxel (2020)).

Nevertheless, setting the expectations bar too high, especially from possibly rushed and flawed small-scale initial clinical trials, which can lead to misleading outcomes, is likely to lead to far more than disappointment in the face of the pandemic.

Quality medical science must be the final arbiter.

## 7. Safe Development of a COVID-19 Vaccine

The viewpoint presented by [12] Lurie, Sharfstein, and Goodman (2020) discusses the development of a provably safe, effective, reliable, timely, affordable, acceptable, approved, and accessible vaccine for the SARS-CoC-2 virus through large-scale clinical trials for a diverse range of test patients is essential to enable society to converge to some form of normalcy in the COVID-19 world.

Any announcements of purported vaccine discoveries should be made only when rigorous testing has been approved by the appropriate medical and healthcare authorities.

Fast tracking approval of vaccines is understandable in uncertain times, but the outcomes of clinical experiments for safe false positives, false negatives, and reinfection risk, among others, must be rigorously and robustly checked according to the highest medical and healthcare regulatory standards in order to mitigate the risk of deploying a vaccine without guaranteed safety standards.

## 8. Dealing with Cancer Grief in COVID-19

The sensitive and moving perspectives by [13] Sanoff (2020) will be taken to heart by anyone who has been diagnosed with terminal and inoperable cancer, or knows someone who has been.

Isolation from society is not unknown for cancer patients, but the added social difficulties arising from COVID-19 can lead to greater struggles in terms of distancing and associated mental health issues.

Quality of life is essential for terminal patients, and oncologists have immense medical and emotional pressure to assist and support terminal cancer patients.

It is providential that oncologists have a mission that focuses on compassion and empathy, among others, and as gatekeepers of the quality of life of cancer patients when they need it most.

When the time comes, cancer patients will miss their family and friends deeply, as well as their oncologist.

## 9. Ranking versus Rating Hospital Quality

Any academic who has been associated with ranking or interpreting ranking systems of individual research output, academic standards of research journals, and associated departments, faculties, and universities, is well aware that measuring quality, efficacy, impact, importance, and influence is based on numerous arbitrary factors that are not necessarily widely accepted.

The same argument applies to annual rankings of departments, faculties, colleges, business schools, and universities in an endeavor to attract prospective students.

Whichever rankings or ratings methods are used, it is critical that any factors which are presented are simple, identifiable, convincing, transparent, balanced, informative, useful, easily interpretable, rigorous, measurable, accountable, reproducible, and, most of all, accurate.

As analyzed in the excellent viewpoint by [14] Bae, Curtis, and Hernandez (2020), hospital rankings and ratings can reduce misinformation in the process of expanding and explaining information about public healthcare delivery outcomes.

It is recognized that there can be significant differences across large geographic areas, financial affordability, socioeconomic environment, racial and ethnic composition, among other pervasive disparities; therefore, any informative ranking or rating system would be more helpful if the healthcare boundaries were localized.

Measuring healthcare quality and outcomes provided by hospitals is subject to numerous arguable and arbitrary factors, such that creating qualitative and quantitative tiers and group ratings might be preferable to numerical rankings that are statistically problematic in the absence of meaningful confidence intervals.

## 10. Effective School Closures during COVID-19

Opening up the economy and society includes essential services and access to education at all levels, which is carefully discussed in this informative viewpoint by [15] Esposito and Principi (2020).

Online learning is feasible at the tertiary level, especially for graduate students, but elementary and secondary (high school) students need encouragement, mentoring, and face-to-face contact to achieve their intellectual potential, especially in a COVID-19 world where isolation can be increasingly restrictive on all members of the community.

With a high likelihood that COVID-19 will not disappear in the foreseeable future, governments and education authorities need to establish strict rules that govern public health safety for school children while they are travelling to and from schools, and activities during school time, including lunches and for any other meals, limiting class sizes, physical distancing, and emphasizing hygiene.

Further extensive testing is required to determine the likelihood of children being diagnosed as COVID-19-positive in school settings, and the attendant unresolved issue of herd immunity.

Classes need not be held every weekday, although requiring school children to be home-schooled is unfeasible due to economic pressures of working parents, lack of knowledge about effective home schooling, and access to online teaching materials, including computer access, especially at short notice.

Substantial information regarding effective infection control, transmissibility, attack rates, and deaths have been made available in the more than two months since the pointed viewpoint was published, but opening up schools remains highly contentious within society.

Comparisons of COVID-19 with other previous coronaviruses such as SARS will not provide an accurate reflection of the future of the global economy, society, or education at all levels.

Informed education public policy is especially important for the uncertain lives of today's children, who must be guaranteed physical, emotional, educational, and social support in their formative years.

## 11. Reducing Exclusions in Clinical Trial Interventions for Delirium

The informative and instructive research message by [16] Martin, DiBlasio, Fowler et al., (2020) provides a rigorous assessment of a mental disorder that generally affects older patients.

Delirium leads to the hospitalization of many older patients, for whom it is not necessarily temporary and reversible, but can lead to greater risks of death and cognitive impairment.

Randomized and controlled clinical trials are subject to arbitrary and unexplained exclusions of at-risk patients from participating, which can affect the conclusions that can be drawn.

A cross-sectional analysis of 89 summary clinical trials data that focused primarily on pharmacologic interventions for prevention, with a reduced focus on treatment, might prove useful for purposes of exploring the aggregate data, generally of adults aged 60+, likely subject to surgery and cared for in ICUs.

Nevertheless, lack of access to anonymous individual data, with general exclusion of patients with a variety of comorbidities and disorders, pre-existing dementia, substance abuse, and advanced or terminal illness, will inevitably restrict the conclusions drawn from being robust for the purposes of predicting the future treatment and behavior of patients.

Excluded older hospitalized patients with a greater risk of delirium also have a range of comorbidities; therefore, expanding the clinical trials to focus on a wider range of patients as reflected in the increasing older patient population will make them more meaningful, while reducing exclusions of clinical trial interventions.

## 12. Weighted versus Unweighted Assessments of COVID-19 Death Inequities

The critical and incisive analysis by [17] Cowger, David Etkins et al., (2020) distinguishes carefully between weighted (or "indirect standardization") and unweighted data in assessing inequities in COVID-19 surveillance and mortality data according to race and ethnicity data provided by the U.S. CDC. The weighted data are given relative to COVID-19 deaths, while the unweighted are given relative to the total population.

If the shares of COVID-19 deaths for each country were proportional to the total population across geographic, racial, and ethnic distributions, there would be no need for weighting.

The publicly available aggregated cross section data were analyzed relative to two population distributions; therefore, unless the two populations are proportional to each other across the cross sections, it is a foregone conclusion that at least one, if not both, of the weighting structures, which were reported as unchanged between 13 May and 7 July 2020, will lead to biased estimates of the unweighted distribution.

This will undeniably bias, upward or downward, the estimated inequities according to race and ethnicity across countries, which will mitigate against correcting the inequities through appropriate allocation of healthcare resources.

The use of such arbitrary weighting densities begs the question as to why such weights were ever considered, let alone used, regardless of the questionable explanatory statement given by the U.S. CDC.

## 13. Strengthening Registered COVID-19 Clinical Trials

The informative research presentation by [18] Pundi, Perino, Harrington et al., (2020) on the characteristics and strengths of registered COVID-19 clinical trials bears serious analysis for designing future clinical trials, and developing relevant and appropriate healthcare policies.

The carefully constructed registered COVID-19 clinical trials data led to the identification and inclusion of 1551 studies for the period 1 March 2011 to 19 May 2020, of which 451 could lead to level 2 evidence, or the highest level of individual study evidence.

The most common primary and secondary outcomes were reported to be mortality, ventilation requirements, and treatment complications.

In addition to the stated limitations, which are required by regulation to include only drug, device, or biological studies to be registered, and with one-half of non-US studies estimated to be non-registered, the following might be considered for future analysis as dealing with the limits to registered COVID-19 clinical trials:

(1)   Expand registration beyond drug, device, and biological studies;
(2)   Broaden the range of drugs and biological compounds, including the topical hydroxychloroquine, remdesivir, antivirals, and corticosteroids;
(3)   Extend the data set beyond 19 May 2020, after which the number of confirmed COVID-19 cases exploded worldwide;
(4)   Analyze the registered COVID-19 clinical trials recursively by day, week, and month as the pandemic progressed;
(5)   Expand blinding, as required for level 2 evidence, and placebo-controlled clinical trials;
(6)   Using clinical trials registered on ClinicalTrials.gov as a benchmark, consider clinical trials that are not registered;
(7)   Consider clinical trials that might be at alternative stages of preliminary development and outcomes;
(8)   Establish alternatives to level 2 evidence beyond placebo-controlled, blinded studies.

Current and impending COVID-19 patients are unlikely to be concerned about some relaxation in strict regulations governing registration and assessment of clinical trials, including satisfying level 2 evidence, as long as safety standards are not unduly compromised in balancing speed, efficacy, usefulness, and the practicality of clinical trials outcomes.

## 14. To Vaccinate or Not in a COVID-19 World

The informative and helpful analysis of vaccination in the COVID-19 world by [19] Rosenthal and Thompson (2020) is essential reading for anyone concerned about the repercussions of the pandemic that is causing havoc to everyone everywhere.

One does not need to be a medical specialist to understand that, although not an immunization panacea for all viral diseases, vaccines can certainly help in preventing spread in the community.

Sensible and sensitive considerations about protecting the most vulnerable members of society, including children, the infirm, and the aged, are essential for conducting diligent and preventive healthcare public policy.

Who would decline the opportunity to access a safe, effective, and affordable vaccine for COVID-19 if such were available?

While the world awaits the discovery of a vaccine to fight the effects of the surging pandemic, it is essential to maintain the highest healthcare standards possible for all members of society, especially those who are most at risk of other viruses that remain, especially when suitable vaccines are available.

Children may not be sufficiently knowledgeable to decide whether they should be vaccinated against a range of preventable illnesses.

If parents are not sufficiently knowledgeable about the consequences of eschewing vaccines based on social, ethnic, cultural, religious, or any other foundation, society has a responsibility to inform, educate, and encourage vaccination.

Ignorance may be bliss, but it is inexcusable and lacks virtue when parents and society fail to protect those who need it most.

## 15. Affordability of COVID-19 Vaccines

The enlightening and informative viewpoint by [20] Shah, Marks, and Hahn (2020) of the U.S. Food and Drug Administration (FDA), regarding an unwavering commitment to satisfying regulatory safeguards for COVID-19 vaccines as foundations for best practice public health, is reassuring.

The FDA, among others, argue that minimizing the risks associated with resurgence and settling safely to a new normal life in a COVID-19 world will require a majority of the population to develop herd immunity against COVID-19.

Targeting herd immunity typically requires the development of safe and effective vaccines, which carries significant financial risk for pharmaceutical companies and even non-profit research organizations because of the high failure rate throughout the development and licensing processes.

It is a statement of fact that safe and effective vaccines are essential to encourage individuals in the population to agree to vaccination.

The viewpoint states that the "FDA has specifically recommended that the primary efficacy endpoint point estimate for a placebo-controlled efficacy trial should be at least 50%, and the statistical success criterion should be that the lower bound is >30%".

Moreover, to achieve widespread immunity through widespread deployment, a vaccine would need to have included racial and ethnic minorities, older adults, and individuals with a range of comorbidities in the clinical trials to reflect the population's characteristics.

What merits explicit mention in any discussion regarding the development and distribution of safe and effective vaccines is the important issue of affordability, especially for the poor and disenfranchised in society.

## 16. Preventing Child Abuse and Maltreatment during COVID-19

The timely and excellent editorial by [21] Greeley (2020) on the prevention of, and protection against, child maltreatment and violence during COVID-19 and at all other times is a welcome addition to the growing literature on the important topic of child abuse and pediatric public healthcare.

Complementary references include, among others, [22] Shekerdemian, Mahmood, Wolfe et al., (2020) on infected children in pediatric ICUs, and [23] Rosenthal and Thompson (2020) on child abuse awareness.

It is a truism that children suffer most from risks associated with inequalities of every imaginable kind, and this sad fact has continued to manifest itself during the COVID-19 pandemic.

Self-isolation, social distancing, quarantining, and lockdowns bring immeasurable stress on all citizens, but children are most vulnerable when they are denied social contact during their formative years.

Any risks to children are compounded when they are exposed to and suffer from physical and emotional maltreatment, harm, abuse, violence, and injury.

Separating children from abusive and criminally negligent parents or other authoritative individuals is essential, but their placement elsewhere, supported by suitable economic and financial resources, must be conducted sensitively and with the welfare of the children given utmost importance.

## 17. Protecting Healthcare Workers from COVID-19 Burnout

The clear and careful dissection by [24] Matsuo, Kobayashi, Taki et al., (2020) of the prevalence of burnout from COVID-19 at an international hospital in Japan is enlightening in drawing attention to the physical, mental, and emotional stress that affects frontline healthcare workers.

The invaluable empirical study was based on an online cross-sectional survey conducted between 6 and 19 April 2020 at a hospital with a large number of COVID-19 patients, namely, St Luke's International Hospital in Tokyo, Japan.

The survey solicited self-reporting responses regarding the demographic characteristics (age and gender) of participants, job category, other risk factors, types of anxiety, and changes compared with before the pandemic, to assess, identify, prevent, and reduce the risk of burnout of frontline healthcare workers.

The sample included 312 respondents, with a median age of 30.5 years, of whom 223 (71.5%) were women, and with median experience of 7.0 years.

Using physicians as the control group, the overall burnout prevalence was 31.4% (98 of 312) compared with the group without burnout, with the primary criterion being high levels of exhaustion.

The investigators acknowledged that limitations of the study included the focus on frontline healthcare workers treating COVID-19 patients in only one hospital, with no baseline burnout level determined before the pandemic.

The key findings from an important and insightful research agenda would be enlarged considerably with more extensive and comprehensive online questionnaires that focus on:

(1)     Extending the cross-sectional sample period beyond April 2020, after which the pandemic exploded;
(2)     Increasing the number of major hospitals in Tokyo and throughout Japan to expand the sample size beyond 312 respondents;
(3)     Including a broader range of healthcare workers according to age, i.e., older than 30.5 years;
(4)     Including a higher proportion of male healthcare workers, up from 28.5%;
(5)     Internationalizing the number of major teaching and comprehensive hospitals across countries;
(6)     Distinguishing between major public and leading private hospitals across countries;
(7)     Reducing the focus on frontline healthcare workers who are dealing primarily with COVID-19 patients;
(8)     De-emphasizing uncertain and possibly unknown and confusing self-assessments before the pandemic;
(9)     Emphasizing different stages of the pandemic, including early, first wave, flattening, and exponentiating second and higher waves;
(10)   Including additional conditioning factors, such as socioeconomic, economic, and financial, to broaden and robustify the empirical findings.

Expanding incisive and informative questionnaires is essential to identify, detect, and predict the factors that enable interventions to reduce and prevent the risk of burnout of frontline healthcare workers, who need protection themselves while administering care and protection to patients who need it most in a world dominated by COVID-19.

## 18. Nursing Home Ratings for COVID-19 Cases

The illuminating research findings by [25] Figueroa, Wadhera, Papanicolas et al., (2020) on healthcare policy and management is instructive for frontline healthcare workers, patients, and carers who are reliant on nursing homes and their associated ratings.

In many countries, a high proportion of deaths from COVID-19 infections have occurred in nursing homes: 27% in the United States.

The informative essay evaluated ratings according to health inspections, quality measures, and nurse staffing, with higher ratings expected to be associated with lower COVID-19 cases.

State-based data from eight state health departments with high COVID-19 cases for the period 1 January to 30 June 2020, were used in the empirical analysis, although the starting point as 1 January 2020 is not explained.

As mentioned in the research report, the quality measures rating is based on the weighted mean of performance across 15 quality measures, and the nurse staffing domain is based on the mean staffing hours per resident by qualified nursing staff.

Three separate ordinal logistic regression models were used to estimate the odds of high-performing facilities, conditional on the number of certified beds and county fixed effects.

The empirical findings indicate that higher nursing home ratings for the nurse staffing domains are associated with fewer COVID-19 cases, although no significant relationship was found for the health inspection or quality measure domains.

The important healthcare policy findings suggest that increasing nursing staff support is critical to reduce the spread of COVID-19.

The report ends with the proposition that more highly rated nursing homes might be better able to test for and diagnose COVID-19, which would lead to higher case numbers.

Conversely, if more lowly rated nursing homes are less able to conduct tests for and diagnose COVID-19, this would lead to lower case numbers.

Further modelling as to whether nursing home ratings are linearly or nonlinearly associated with more rigorous testing regimes for COVID-19 would be instrumental in determining the strength of association between increasing nursing staff support and the number of COVID-19 cases.

## 19. Challenging Oncology Practice during COVID-19

The intriguing viewpoint by [26] Gyawali, Poudyal, and Eisenhauer (2020) on oncology highlights the sensitive balance required between best oncology practice and the commingling effects of COVID-19.

Reducing the number of physical follow-up visits and tests through electronic or telehealth alternatives is beneficial to cancer patients and healthcare providers, especially the latter, who have numerous additional concerns about appropriate cancer therapies in the presence of the pandemic.

Patient healthcare involves determining advanced cancer therapies through education in a technical and demanding arena, especially when physical face-to-face contact is replaced by Zoom or Skype interactions in order to reduce the likelihood of serious complications for cancer patients.

## 20. Addressing Racial, Ethnic, Social, and Economic Divides in COVID-19

The informative viewpoint by [27] Galea and Abdalla (2020) on the racial, ethnic, social, and economic divides arising from COVID-19 is even more striking almost two months after its publication.

With almost 5 million total cases and close to 160,000 deaths from COVID-19 as of 4 August 2020 (https://www.worldometers.info/coronavirus/), the data show that the pandemic is exponentiating rather than flattening.

Society has ground to a halt in many cities, states, provinces, regions, countries, and continents and, where it has not, the number of cases is experiencing a second wave, where the most disadvantaged in society face greater risks of contracting COVID-19, and dying from it.

Unemployment is rife, companies and jobs are disappearing, racial, ethnic, and civil unrest are increasing in many countries, and police and other frontline emergency workers are facing increasing stress.

Consequently, societies are undergoing drastic transformations that have not been seen since the greatest pandemic in recent history, the 1918–1920 Spanish flu, let alone the Great Depression of 1929–1933.

Tragic extraneous structural changes provide opportunities for governments to conduct essential changes in public policy that, in turn, can affect and correct entrenched economic, racial, ethnic, and social inequities.

Entrenched unemployment and extreme economic inequities are difficult to correct, but governments have an inviting opportunity to rebalance unemployment benefits, economic investment, welfare payments, and increased access to healthcare facilities through appropriate legislation.

Economic, racial, ethnic, and health inequities are inherently interdependent; thus, legislation targeting specific inequities, such as unbalanced and racist employment practices, restricted access to medical and healthcare facilities, and reduced life expectancy, are required.

Racial, ethnic, and economic divides in civilized societies are unacceptable at any time, but positive side-effects of the COVID-19 pandemic are the opportunity and imperative for progressive administrations to engage in proactive public policy decision making for the common good.

## 21. Violence against Sexual Minority Adolescents

The detailed, enlightening, and novel clinical analysis by [28] Caputi, Shover, and Watson (2020) on physical and sexual violence among sexual minority, specifically LGBQ, adolescents in the United States is disturbing in terms of public health hazards, as well as the criminally liable behavior of the perpetrators of such violence.

Public information on the prevalence of physical and sexual assault against sexual minority adults is generally available through many studies, but not for sexual minority adolescents.

Consequently, the findings in the paper provide an urgent wake-up call for researchers, public policy decision makers, clinicians, and pediatricians to develop interventions to reduce the risks of violence committed against sexual minority adolescents.

The pooled data were accessed from the National Youth Risk Behavior Survey (YRBS), conducted every 2 years by the U.S. Centers for Disease Control and Prevention (CDC), for 2015 and 2017, and are based on anonymous self-assessments.

The questionnaires required self-identification of sexual orientation as lesbian, gay or bisexual, with "not sure" interpreted as queer or questioning (that is, still exploring one's sexual orientation), for the initialism LGBQ.

Transgender and intersexed (that is, with ambiguous biological sex characteristics) were not included in the questionnaire, which is surprising, because transgender (although not intersexed) has been included in LGBT since Gallup started tracking the percentage of U.S. adults identifying as LGBT every two years in 2012 (https://news.gallup.com/poll/234863/estimate-lgbt-population-rises.aspx).

A direct comparison between adult LGBT (3.9% in 2015 and 4.5% in 2017) and adolescent LGBQ (12.9% based on pooled data for 2015 and 2017) is not strictly accurate, but is nevertheless instructive in terms of the apparent large differences between adult and adolescent self-identification.

The important questionnaire conducted by the U.S. CDC should expand the sexual minority classifications to include LGBTIQ adolescents to enable more accurate, targeted, and robust public health interventions against unacceptable and elevated physical and sexual violence.

## 22. Primary Tumors, Metastasis, and Stages of Cancer

The clear and instructive explanation by [29] Patel and West (2020) on oncology and healthcare professionals of the I–IV stages of cancer in terms of treatments and possible cures is immensely helpful to cancer patients and their carers in terms of understanding and dealing with cancer.

Measurement of cancer in terms of tumor size and location are crucial in cancer treatment, according to the primary tumor, and the extent and speed of spread to other organs.

The definitions of the I–IV stages can differ according to the types of cancer, which seem to be determined by the primary tumor.

What does not seem to have been stated explicitly is whether advanced cancer that has metastasized might also be defined as having different stages.

Defining the stages of primary and metastasized tumors can help to guide patients and carers using best practice cancer management strategies.

## 23. Reforming Cancer Clinical Trials during COVID-19

The instructive and illuminating viewpoint by [30] Nabdan, Choueiri, and Mato (2020) highlights the reform and execution of clinical trials using novel therapies, advanced surgical techniques, radiotherapy, and supportive care for cancer patients.

In the absence of physical or psychological impediments, it is difficult to understand why fewer than 10% of adult patients with cancer in the United States enroll in clinical trials.

The experts state that the emergence of COVID-19 has created a barrier to entry for enrolling in clinical trials, but it also creates opportunities for instigating reforms.

Such reforms include the emergence of telemedicine and virtual visits, reducing administrative paperwork, relaxing protocols, localizing laboratories to evaluate the outcomes of clinical trials, education of the broad treatment population, reducing the number of patients required for approved clinical trials, decreasing the frequency of imaging studies, and patient healthcare involving advanced cancer therapies through education in a technical and demanding arena.

Innovative and flexible reforms are particularly important when physical face-to-face contact is replaced by Zoom or Skype interactions in order to reduce the likelihood of serious complications for cancer patients.

Critically ill cancer patients do not need progression-free survival as an end point, but rather quality of life and informed consent about survival arising from the best practice clinical trial methodology in the COVID-19 era.

## 24. Supporting Carers, Protection, and Prevention during COVID-19

The sensitive and caring advice and precautionary recommendations by [31] Mortenson, Malani, and Ernst (2020) on patient care and preventing household spread are revealing and informative for purposes of protecting carers and their patients who are under isolation, distancing, quarantine, or lockdown.

Isolation leads to severe difficulties in dealing with serious illness, including mental illness; thus, patients and their carers need to support and protect each other in order to prevent infection, with a high degree of hygiene being essential.

Informed careers are needed to care effectively for patients with both mild and severe illness, especially because there is substantial information that can be useful or misleading, especially in the news media and social media.

Fortunately, most carers are up to the task when their kindness and thoughtful care and consideration are needed the most, for which patients are eternally grateful.

## 25. Knowns and Unknowns of Benign and Malignant Tumors

The helpful and informative explanation by [32] Patel (2020) on benign versus malignant tumors, or an abnormal mass of cells in the body through division or not dying, is instructive for all concerned, especially patients and their carers.

The classification as benign or malignant tumors depends on multiple characteristics, including regular versus irregular borders, the speed of invasion of surrounding tissues, and the possible invasive spread to other bodily parts through metastasis.

Although benign tumors are unlikely to recur after removal, specific types of benign tumors can become malignant, although it is not stated unequivocally whether the change from benign to malignant refers to primary, secondary or tertiary level tumors.

It is known that malignant tumors are cancerous through metastasis from the original source to other internal organs, most frequently, the liver, lungs, brain, and bone, although the specific order and sequence of the secondary and tertiary stages are not necessarily known.

It appears to be known that malignant tumors will remain malignant as they spread to other organs at the secondary and tertiary stages.

The possibly known degree of malignancy would be useful information for the understanding of malignant tumors and cancers, as would the likelihood of survival for specified durations at the primary, secondary, and tertiary stages.

Such information and associated recommendations at each metastatic stage would be assuring to cancer patients and their carers.

## 26. Protection of Healthcare Personnel against COVID-19

The insightful research letter by [33] Moscola, Sembaiwe, Jarrett et al., (2020) on the prevalence of SARS-CoV-2 antibodies in healthcare personnel, who are exposed to

the dangerous and predatory COVID-19, is essential reading by everyone involved in healthcare, as well as the patients, carers, and family of those who are infected with COVID-19.

Broad-based access to testing for the disease is essential for healthcare workers for self-protection, as well protecting their entirely dependent patients.

A straightforward free and voluntary antibody test is available to all healthcare personnel, who were analyzed during the period 20 April 2020 to 23 June 2020, according to demographics, primary work location and type, and suspected exposure to the SARS-CoV-2 virus.

The authors explain that seroprevalence with 95% confidence interval was calculated by the exact binomial technique, and associations among seroprevalence and a set of conditioning variables were assessed using Poisson logistic regression.

As of 23 June 2020, 46,117 healthcare personnel had been tested by reverse transcriptase–polymerase chain reaction (PCR), with a median age of 42, 73.7% female, 0.8% multiracial, 28.4% nurses, and 9.3% physicians.

Overall, 13.7% of volunteer healthcare personnel were seropositive, whereas 34.8% (93.5%) of those with previous PCR testing were PCR-positive (respectively, seropositive), and 9.0% with no PCR testing were seropositive.

In summary, high levels of suspicion of exposure to the virus, and prior positive PCR test outcomes, had the highest relationship to seropositivity.

The results of the study would have been even more meaningful and robust with:

(1)    A broader representation of males and multiracial;
(2)    Proportional representation of physicians to nurses;
(3)    Distinguishing between male and female nurses, and male and female physicians;
(4)    Duration between prior positive PCR tests and subsequent PCR positivity and seropositivity.

Robust findings from voluntary participation are essential for healthcare personnel to be safe and healthy, as well as everyone with whom they come into close personal contact, who need their protection the most.

## 27. Comparing Excess Deaths in NYC for the 1918 Influenza Pandemic and COVID-19

This important and innovative research finding by [34] Faust, Lin, and del Rio (2020) is a significant contribution to knowledge of the estimated excess deaths between the most alarming pandemics over the past 102 years, although the endpoint for COVID-19 is nowhere in sight.

In order to contextualize the invaluable findings, the 1918 flu pandemic, which lasted from 1918 to mid-1920, is the most devastating in recorded history, occurred in a world with a population of around 1.8 million, and led to an estimated upper bound of 50 million deaths globally, with the most severe months being the fall of 1918.

An estimated upper bound of 675,000 died in the United States, which had a population of 103 million in 1918, compared with 331 million in 2020.

Life expectancy in the United States was 47.2 years in 1918, 55.3 years in 1919, 55.4 years in 1920, and 78.9 in 2019 (https://ourworldindata.org/spanish-flu-largest-influenza-pandemic-in-history) (by Max Roser, posted 4 March 2020).

The study compares excess deaths in the peak of NYC (October–November 1918) with the initial stages of COVID-19 in NYC (March–May 2020), which includes the peaks for COVID-19 which seem to have occurred on 8 April (799 deaths), 10 April (783 deaths), 13 April (778 deaths), and 5 May (952 deaths) (https://www.google.com/search?ei=TUM3×7 f9D7-X4-EP6_2_yAw&q=numbers+of+deaths+in+NYC+from+COVID-19&oq=numbers+of +deaths+in+NYC+from+COVID19&gs_lcp=CgZwc3ktYWIQAzIICCEQFhAdEB4yCAghEB YQHRAeULQSWNkdYMNKaABwAHgAgAHoAYgByA6SAQUwLjQuNZgBAKABAaoBB 2d3cy13aXrAAQE&sclient=psy-ab&ved=0ahUKEwj318b1j5zrAhW_yzgGHev-D8kQ4dUD CAw&uact=5).

Important caveats in comparing the two pandemics include improvements in hygiene and healthcare advances over the past century.

However, these limitations are not factored directly into the estimates of excess deaths of the two pandemics, which suggest that mortality from COVID-19 might be greater than the 1918 flu epidemic.

It is worth highlighting that the speed of transmission via the incubation period and serial interval for successive cases are shorter for influenza than COVID-19, the reproductive number is higher for COVID-19 than influenza, and children are regarded as drivers of influenza although not COVID-19 (WHO, Geneva, Switzerland, https://www.who.int/emergencies/diseases/novel-coronavirus-2019/question-and-answers-hub/q-a-detail/q-a-similarities-and-differences-COVID-19-and-influenza?gclid=EAIaIQobChMIlPvp0pSc6wIVT38rCh0wZg_HEAAYASAAEgJlTPD_BwE).

These differences, among others, might be factored into follow-up studies that would shed further light on the likely future impact of COVID-19, which seems to have passed its peak in NYC, though not nationally.

If the findings from this informative study can be extended to other major metropolitan centers throughout the United States, this would suggest that excess deaths from COVID-19 cannot be compared with the seasonal flu, but only with a once-in-a-century flu pandemic.

## 28. Is a Full Recovery from COVID-19 Possible?

The authors [35] Carfi, Bernabei, Landi et al., (2020) are to be commended for their informative and instructive message about persistent symptoms from acute COVID-19, based on 179 patients for the period 21 April to 29 May 2020.

Are persistent symptoms an indication of a dormant SARS-CoV-2 virus that can be excited to deliver recurrence of the disease, or mistaken diagnoses that are undetected using tests for COVID-19 that lack power?

A question that does not yet seem to have been considered as seriously as is warranted is the following: Is it possible to have a full recovery from COVID-19?

## 29. Mental Health Impact of COVID-19 on Young Adults

The informative message from [36] Stephenson (2020) has raised awareness of the important mental health issues that have arisen in the COVID-19 era.

The behavior of young adults, in particular, is of concern because they seem to be the most highly stressed cohort of adults.

The reasons can be diverse, but obvious explanations are:

(1) Difficulties in finding gainful employment as the economy shuts down in many countries where there are national, regional, state, and provincial lockdowns;

(2) Lack of experience and patience in dealing with new social norms that include observing, respecting, and practicing social distancing, self-isolation, quarantining, and mandated lockdowns.

Experiencing stress and incessant anxiety about issues young adults cannot control is a recipe for mental health issues.

## 30. Ranking and Rating Nursing Home Quality

As discussed informatively by [25] Figueroa, Wadhera, Papanicolas et al., (2020), there are significant difficulties in establishing any qualitative and quantitative ranking or rating system based on what might be argued to be arbitrary criteria, as discussed informatively by experts on healthcare policy.

Similar issues have been raised and discussed extensively by [14] Bae, Curtis, and Hernandez (2020), as presented in Section 9 above.

A useful suggestion that was raised in the context of hospitals is that "creating qualitative and quantitative tiers and group ratings might be preferable to numerical rankings that are statistically problematic in the absence of meaningful confidence intervals".

Extension of the argument to nursing homes is apropos.

### 31. Charles Dickens and Cancer Outcome Disparities in the Time of COVID-19

In *A Tale of Two Cities* by Charles Dickens, the most memorable lines are the first and last sentences, which resonate with disparities in cancer outcomes, as analyzed in the instructive viewpoint by [37] Balogun, Bea, and Phillips (2020) on medical and cancer treatment during the COVID-19 pandemic, and have been immortalized as:

"It was the best of times, it was the worst of times, it was the age of wisdom, it was the spring of hope, it was the winter of despair."

and

"It is a far, far better thing that I do, than I have ever done."

The disparities in the time of Dickens were essentially social and economic inequities.

In the time of COVID-19, the socioeconomic disparities are extended to include higher unemployment rates, underserved ethnicity, race, and geographic neighborhoods, especially for minority populations, an inability to work from home, supervising children who are required to be homeschooled, lack of access to telemedicine, and the lack of understanding of, and access to, healthcare and minimal insurance coverage, derived in several different parts of New York City, the United States, and China.

Such socioeconomic disparities also have significant effects on known pre-existing cancer treatment disparities, although the extent and devastation of COVID-19 on the magnification of such disparities is not yet known.

An important unknown issue is how various types of active or prior cancer might interact with COVID-19.

Crucial known unknowns that are essential for proactive public healthcare policy, including oncology, are:

(1) Examining whether the causal impact is from COVID-19 to cancer, or if there might be reverse causality from cancer to COVID-19;
(2) The different rates of susceptibilities based on malignant primary, secondary, and tertiary cancers;
(3) The rates of reinfection from COVID-19 after infected patients have tested negative;
(4) The durations between initial diagnosis and clearance with negative test outcomes, and reinfection from COVID-19;
(5) Whether the predatory COVID-19 disease ever leaves the body completely, but may lie dormant to resurface at an unknown, unexpected, and alarming length of time after a negative diagnosis.

The reference by the prescient Dickens in the last sentence of his novel is alarming in relation to the behavior of a number of senior administrative officials in the time of COVID-19:

"the period was so far like the present period, that some of its noisiest authorities insisted on its being received, for good or for evil, in the superlative degree of comparison only."

### 32. Schooling Children in the Time of COVID-19

Education and personal safety are essential for every member of society, but all the more so for children in their formative years.

The incisive editorial by [38] Dooley, Simpson, and Beers (2020) provides an excellent overview of challenging and pressing issues, with an emphasis on the lack of equity, in a climate that is clouded by conflicting information regarding the COVID-19 pandemic, safety in opening up schools, and deeply divisive political considerations.

Inequalities in income and employment are reflected vividly in educational inequality, especially according to race, ethnicity, and medical and physical disabilities, frequently at underfunded schools with inadequate staff levels, and where homeschooling using best practice computer facilities and access to online educational information are not serious options for a substantial proportion of the population, especially in single-parent households, where staying at home to engage in home education is not feasible.

It is well known that parents of Black, Latinx, and Native American students are more likely to be denied access to flexible employment practices, through socioeconomic and educational inequality, and lack access to the best educational and healthcare facilities.

Students from disadvantaged backgrounds also tend to suffer from higher levels of stress and anxiety, which affects academic progress in a vicious cycle, especially in the time of COVID-19.

Until such time as socioeconomic, educational, and health inequality issues are addressed to target and resolve the underlying endemic problems, including protecting children, as well as their teachers and carers, from the prevailing pandemic, opening up schools will continue the discrimination against large segments of the population, which reflects on a society that permits such inequality to exist.

## 33. Duration of Isolation after Positive COVID-19 Test Outcomes

As discussed by [39] Stephenson (2020), it is unarguably essential to revise and update important information and guidance regarding the estimated optimal duration of isolation after a positive test outcome for COVID-19.

However, changing messages can lead to confusion, especially to those who are not well versed in interpreting possibly confounding recommendations and guidance.

Is a shorter or longer period required for isolation, i.e., 10 days, 20 days, or even more, based on the lower and upper bounds derived from statistical testing, or interpreting the available descriptive statistics?

Clear and accurate guidance is particularly important for young adults, whether immunocompromised or not, who are known to be highly susceptible to stress and mental healthcare issues.

The reality of prolonged illness, especially for older patients with or without multiple chronic medical conditions, begs the question of the chances of full recovery from COVID-19 for most patients, and how long it might take.

## 34. Stress, Anxiety, and Mental Health during COVID-19

As discussed by [35] Carfo, Bernebei, Landi et al., (2020), persistent symptoms after patients have been diagnosed with COVID-19 are of deep concern, especially if the endpoint is unknown.

Associated with the disease itself, as well as the parallel issues of a socially distanced society and closed economy, are the increased stress, anxiety, and severe mental health issues experienced across all cohorts, but especially for the young and those who suffer socioeconomic inequity and disadvantage.

Mental health issues are almost certain to worsen as the unknown endpoint of the pandemic stretches well into the future.

## 35. Ethical Clinical Trials for a COVID-19 Vaccine

The detailed and informative recommendations presented in the excellent viewpoint by [40] Wang, Zenilman, and Brinkley-Rubinstein (2020) regarding ethical considerations underlying clinical trials for a COVID-19 vaccine by medical and health experts need to be embraced by all who are involved in approving or conducting experimental trials, or are intending to do so.

Excluding certain regimes worldwide where participation in experimental clinical trials is anything but voluntary, as well as correctional facilities in many countries, voluntary participation in clinical trials is essential in the development of a safe and effective vaccine against COVID-19.

Informed consent is essential for voluntary participation; otherwise, the clinical trials are coercive and exploitative, and hence, unethical.

During a pandemic, those who are at the coalface need, and can benefit immediately from, improved healthcare, while participating in clinical trials for a vaccine, as well as

aftercare, both of which might be at a superior level compared with the underfunding of healthcare needs at correctional facilities, except during clinical trials.

Such considerations relate directly to individuals in correctional facilities, who may also suffer from a range of comorbidities, which might be undergoing treatment, or not.

Coercion is not involved if infected and healthy individuals incarcerated in correctional facilities are selected voluntarily, randomly, and proportionately, relative to the overall nonincarcerated population, in terms of race, ethnicity, and pre-existing physical and mental health conditions.

Ethical considerations are established to protect participants in clinical trials, but the general population, incarcerated or not, also needs protection through the discovery of a safe, effective, and affordable vaccine.

A sensible and sensitive balance of ethical and practical considerations is required in a timely manner to deal with the COVID-19 pandemic that permeates all levels of society.

## 36. Oxygenating Severe Respiratory Patients with COVID-19

The informative research letter by [41] Mustafa, Alexander, Joshi et al., (2020) on oxygenating patients with COVID-19 and extreme respiratory failure is concerned with easing the burden on patients who require prolonged mechanical ventilation, which may not always succeed.

The need to extend ventilation is associated with sedation and immobility, which is an added burden to the possible long recovery time for patients.

Extended periods of immobility affect bodily functions, and can lead to morbidity, disability, and mortality, all of which need to be mitigated in healthcare risk management strategies.

A small sample of 40 patients in Chicago from 17 March to 17 July 2020, with a primary outcome being survival, resulted in encouraging and promising findings from the innovative study.

Further studies with a larger cohort of patients with different socioeconomic, racial, ethnicity years of age, obesity, and non-survival rates, with varying degrees of respiratory failure and exposure to COVID-19, would be revealing about the combined recurrence of COVID-19 and severe respiratory failure, leading to further periods of ventilation and oxygenation, morbidity, disability, and mortality, while adding depth to the current evidence.

## 37. Palliative Care during COVID-19

As discussed by [42] Cooper and Bernacki (2020), palliative care is undoubtedly essential for patients suffering from debilitating and untreatable severe illness, but might be problematic for those who are undergoing treatment for ongoing illnesses, including cancer.

Being interviewed for purposes of palliative care at the beginning of chemotherapy treatment of unknown duration is not necessarily reassuring, regardless of the best intentions of physicians and other healthcare workers, whose views should always be considered, as should those of the patient.

## 38. Loss of Physicians by Suicide

The commentary by [43] Danhauer, Files and Freischlag on the loss of physicians by suicide emphasizes the extreme stress, anxiety, and mental health issues faced by those who are needed the most.

The increasing pressure on all members of society facing social distancing, self-isolation, quarantining, and lockdowns are presently being examined by psychiatrists and mental healthcare professionals, for all age cohorts in the population.

Emphasis and immediacy should be directed to the physical and mental well-being of physicians, who face increasing risk of extreme health pressures, especially as the COVID-19 pandemic continues to spread.

## 39. Concluding Remarks

The highly infectious, contagious, and rapidly mutating SARS-CoV-2 virus that causes COVID-19 has wreaked havoc on the global community in terms of every known and imaginable parameter. The COVID-19 pandemic has highlighted the lack of preparation by the World Health Organization (WHO) and every national government worldwide to deal with such unpredictable structural changes in an increasingly volatile world.

This paper has provided some perspectives on 44 interesting and highly topical research papers that have been published in the *Journal of the American Medical Association (JAMA)*. The diverse topics, including public health, general medicine, internal medicine, oncology, pediatrics, geriatrics, and biostatistics, demonstrate that there are more unknowns than knowns regarding COVID-19.

The aggressive developments of COVID-19 are exploding on a daily basis, and the curve of the number of total cases is exponentiating rather than flattening in virtually every country worldwide.

It seems that mutations are also expanding rapidly; therefore, if and when a vaccine is discovered, it is not likely to be as effective for the many waves of the mutated viruses.

The world has seen nothing yet.

**Supplementary Materials:** The following are available online at https://www.mdpi.com/article/10.3390/sci3040038/s1, Review Reports 1–4.

**Funding:** This research received no external funding.

**Institutional Review Board Statement:** Not applicable.

**Informed Consent Statement:** Not applicable.

**Data Availability Statement:** Not applicable.

**Acknowledgments:** The author is most grateful to the Academic Editor for very helpful comments and suggestions. For financial support, the author acknowledges the Australian Research Council and the Ministry of Science and Technology (MOST), Taiwan.

**Conflicts of Interest:** The author declares no conflict of interest.

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
