# Peer review of "Perspectives on Topical Medical Research in the COVID-19 Era"

_sci, doi:10.3390/sci3040038_

Round 1

Reviewer 1 Report

The purpose of this paper is to provide some perspectives on 39 interesting and highly topical research papers that have been published in JAMA, at the time of writing, within the past two weeks.

There are a lot of keywords.

In the introduction, there is a linked website within the text. Can be inserted as a normal citation. 

The author used two forms of citations [number] and Name (year). It must provide a unique way presentation.

By the way, it is not clear why not provide a reference to other relevant journals and papers. Still, this paper just provides a systemic readable JAMA paper. 

Reviewer 2 Report

The articles were referenced at the beginning of mini review for those articles. The conclusions from those articles followed initial introduction. It might be better to state that these are original authors’ conclusions or provide reference for those conclusions. Not restating that these are conclusions from original authors in some sections, might appear that these are conclusions from the author of this review and it can cause some sort of misinformation.

Reviewer 3 Report

This paper reads more like an editorial, series of opinions, or annotated bibliography than a peer-reviewed research paper.

Readers looking for an overly dramatic snapshot of some papers, most of which are about COVID-19 or cancer, published a few months into the pandemic, are in luck. 

It is a series of brief summaries of other papers published in JAMA journals in 2020, relatively early in the COVID-19 pandemic. The title suggests that they will all be COVID-related, but many are not. Cancer is also a theme.  Nearly each paper has a brief section dedicated to that paper; although some papers cover similar topics, their discussions are not grouped, which would have been more sensible. Nearly all sections begin with complimentary, excessive praise, such as "The informative and instructive research message..." (line 257) or "The clear and candid dissection..." (line 398) - these rarely actually inform the reader. Similarly, the paper has several throwaway statements like "Quality medical science must be the final arbiter." (line 175). Excessive descriptions like "... SARS-COV-2 … has wreaked havoc on the world community in terms of every imaginable measure" (lines 13 and 36-37) detract from the message of the paper. Many of the so-called "keywords" are long phrases that are simply borrowed from the text.  

The author makes numerous unreferenced assertions about matters related to the paper - for example, Section 10, beginning on line 228, contains several unreferenced assertions about children and about education.  There are many more.

I could not help reacting to the following rhetorical question (lines 341-342): "Who would decline the opportunity to access a safe, effective, and affordable vaccine for COVI-19 if such were available?" - it turns out, of course, that many people would decline it!

Reviewer 4 Report

I cannot understand what the author asserted in this manuscript. Moreover, the author quoted 44 papers that have been published in JAMA at the time of writing within the past 2 weeks, with citing 1 or 2 papers in each section. These references were not enough to assert your mentions. Importantly, we have experienced first and second waves of COVID-19. Therefore, I suggest that the differences of perspectives are mentioned in first and second waves of COVID-19 using more papers.

  1. Some sections (5, 11, 19, 21, 22, 25, 26, 31, 34, 38, 39) seem not to be associated with COVID-19.
  2. Strengthening resistered COVID-19 clinical trials: What is the definition of level 2 evidence?